# ConfRAG: Confidence-Guided Retrieval-Augmented Generation

## ABSTRACT

*Can Large Language Models (LLMs) be trained to avoid hallucinating factual statements, and can Retrieval-Augmented Generation (RAG) be triggered only when necessary to reduce retrieval and computation costs?* In this work, we address both challenges simultaneously. We introduce CONFQA, a fine-tuning strategy that reduces hallucination rates from 20–40% to below 5% across multiple factuality benchmarks. The approach is simple: when the model answers correctly, it is trained to output the answer; otherwise, it is trained to respond with "I am unsure." Two design choices make this training effective: (1) a dampening prompt ("answer only if you are confident") that explicitly discourages overconfident hallucinations, and (2) training data drawn from atomic factual statements (e.g., knowledge graph attribute values), which calibrates model confidence and yields robust generalization across domains and question types. Building on CONFQA, we propose CONFRAG, a triggering strategy that invokes RAG only when the model responses with unsure. This framework achieves accuracy above 95% in ideal case while reducing unnecessary external retrievals by over 30%.

## 1 INTRODUCTION

Despite the remarkable capabilities that Large Language Models (LLMs) have demonstrated, *hallucination of factual statements* remains a challenge (Maynez et al., 2020; Zhou et al., 2021; Ji et al., 2023). It has been broadly realized that factual information shall not be *fabricated* or *generated*, instead shall be anchored in *internally parameterized neural* knowledge or *externally recorded symbolic* content (stored in knowledge graphs, webpages, or other repositories). Significant progress has been made in both knowledge internalization through pre-training (Grattafiori et al., 2024) and external knowledge utilization via Retrieval-Augmented Generation (RAG) (Wei et al., 2021; Yu et al., 2022; Gao et al., 2024; Fan et al., 2024; Huang & Huang, 2024). However, a critical question remains: *when should LLMs rely on parameterized knowledge versus external sources?*

Existing RAG-triggering strategies tend to fall in three categories. Industry practice typically relies on manual or coarse-grained triggers—for instance, enabling RAG only when a user selects a particular model version, when the query falls into certain domains, or when the requested information is known to change over time. However, as highlighted in Head-to-Tail (Sun et al., 2023a), LLMs can still make mistakes even when answering static questions, especially for entities of torso-to-tail popularity, and such errors happen across domains. In contrast, in academic settings, triggering is often decided at the token level, for example when a generated token exhibits high entropy, high self-reported attention, or low confidence (Su et al., 2024; Jiang et al., 2023). This approach requires close monitoring of hidden-state signals derived from LLM's intermediate activations, thus not always practical. There are also approaches that *always* trigger RAG, and decide afterwards whether the retrieved content are relevant, sufficient, and superior to internalized knowledge (Li et al., 2025), or using *prompt* based methods to instruct LLM to output uncertainty and triggers RAG when LLM outputs uncertain (Ni et al., 2024).

In this paper, we focus on the problem of determining *when to trigger RAG for questions that seek static information*. The underlying intuition is straightforward: if an LLM can accurately assess its own knowledge, it should only consult external sources when it recognizes uncertainty. However, our experiments on three benchmarks confirm that self-reported confidence is systematically overestimated (align with observations in Wei et al. (2024a); Xiong et al. (2024)) and therefore unreliable

for RAG-triggering decisions (Figure 3). To address this, we explicitly teach LLMs to estimate their confidence in factual responses and use this calibrated confidence as the basis for triggering RAG. In particular, we make the following three contributions.

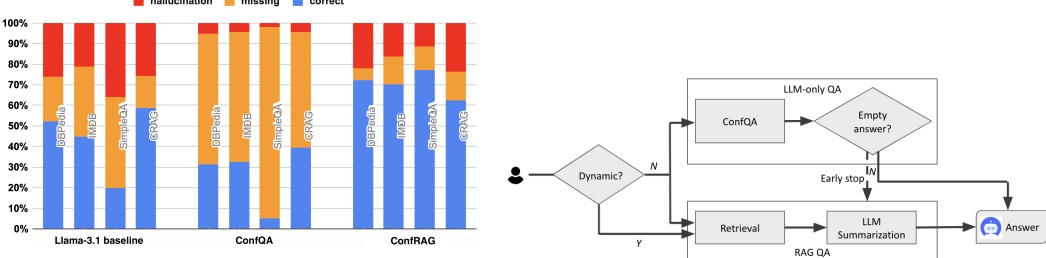

Figure 1: Factuality improvement of CON-FRAG and fine-tuned model CONFQA.

Figure 2: CONFRAG invocation architecture.

Our first contribution is a RAG triggering strategy called CONFRAG, which formulates the triggering decision as detecting low-confidence answers (see Figure 2 for the architecture). For static factual questions, the framework runs LLM-based QA and a RAG pipeline in *parallel*, but halts the RAG process early if the LLM responses with a positive answer rather than "I am unsure". Unlike approaches that rely on inspecting hidden-state signals (Su et al., 2024; Jiang et al., 2023), or make the decision afterwards based on retrieval content (Li et al., 2025), or making decision based on prompt instructed certainty response of LLM (Ni et al., 2024), our proposed strategy is lightweight, low latency, more accurate and broadly applicable.

Our second contribution is an effective fine-tuning method for confidence recognition, called CON-FQA, which forms the core of our triggering strategy. Our training procedure checks the LLM's inherent answer to a question, and teaches it to state "I am unsure" when the answer is incorrect. Though seemingly simple, two key design choices distinguish it from existing work (Zhang et al., 2024a; Cohen et al., 2024; Cheng et al., 2024; Yang et al., 2024b; Kapoor et al., 2025) and make it effective. First, we introduce a *dampening* prompt—*"Answer only if you are confident"*, which plays a crucial role in shaping the LLM's behavior. Second, the training data comprise exclusively of *simple* factual questions about entity attributes; by focusing the regularization on such atomic facts, which serve as building blocks for more complicated statements, the training enables LLMs to generalize this confidence behavior to broader domains.

Our final contribution is a comprehensive empirical study across 7 benchmarks spanning short-form QA, long-form QA, and general knowledge QA. We show that, in theory, CONFRAG can achieve over 95% accuracy with a perfect RAG system; with a real RAG system, CONFRAG attains QA accuracy comparable to always invoking RAG, while reducing P50 latency by over 600ms on CRAG (Yang et al., 2024a). Our experiments also reveal two standalone uses of CONFQA when RAG is not yet an option (such as in Speech-in Speech-out systems (Xie & Wu, 2024; Nguyen et al., 2022)): for maintaining correctness with reduced hallucinations, we recommend CONFQA *without* the dampener in inference, which preserves accuracy with a mild reduction in hallucinations; for minimizing hallucinations, we recommend CONFQA *with* the dampener, which reduces hallucinations to under 5% (see Figure 1).

## 2 RELATED WORK

There are three bodies of work related to our work: RAG triggering, LLM confidence measurement, and hallucination suppression. We next discuss each in detail.

**RAG triggering:** RAG has been extensively researched in academia and widely applied in industry to enhance LLMs' capability to answer factual questions accurately; there have been numerous surveys on RAG (Wei et al., 2021; Yu et al., 2022; Gao et al., 2024; Fan et al., 2024; Huang & Huang, 2024). What is closely related to our work is RAG triggering strategies. The first class of strategies use *token-level* confidence, and often need multiple retrievals sequentially. Su et al. (2024) propose a strategy to trigger RAG when the generated token has high entropy or high self-reported

attention. Similarly, Jiang et al. (2023) propose to use LLM internalized knowledge for generation, and apply RAG when the confidence of the next token is low. Peng et al. (2023) proposes a system that conducts retrieval and answer generation iteratively, revising the LLM prompts to improve model responses using the factuality score from an automatic verifier. Self-RAG (Asai et al., 2023) trains a model to adaptively retrieves documents and do self-reflection using special tokens.

The second class of strategies always trigger RAG, but improve LLM's awareness of uncertainty with retrieved documents in answer generation. Li et al. (2025) fine-tune LLMs to acknowledge uncertainty when provided documents are insufficient. (Ma et al., 2024) propose reducing in-context hallucination when retrieval results cannot support the generated claim, or when LLM fails to use the retrieval results to generate a correct answer.

The third class of strategies using prompt to instruct LLM to express uncertainty in its response. And then decide when to trigger RAG based on the uncertainty. Ni et al. (2024) proposes different prompt templates and multi-step reasoning to decide when to leverage RAG for generation. Note this could also be considered as an example of the LLM confidence measurement work discussed in the following paragraphs.

Our work focuses on *fact-level* confidence, can apply in situations when token-level confidence is unavailable. It does not require retrieval-first, thus can save resources and latency. It relies on fine tuned model, thus is better on factuality metric compared with prompt based method and thus improves overall triggering accuracy and F measure.

**LLM confidence measurement:** This idea of teaching LLM models to be aware of its uncertainty and understand its knowledge boundary dated back to pre-LLM time (Mielke et al., 2022; Kadavath et al., 2022; Lin et al., 2022). Later works compare self-reported confidence and answer accuracy (Xiong et al., 2024), define metrics to better measure LLMs uncertainty (Kuhn et al., 2023), prompts LLMs to state its calibrated confidence (Tian et al., 2023b), relies on consensus between multiple LLM generations (Yadkori et al., 2024) or among multiple LLMs (Feng et al., 2024; Zhang et al., 2023), and uses local intrinsic dimension (Yin et al., 2024). Chen et al. (2025) extends the idea to unsupervised learning by proposing a two-stage training process called CoKE. ConfQA applies fine-tuning to obtain better results in predicting confidence.

The works closest to ours fine-tune LLMs to better calibrate its confidence while providing an answer, and instruct LLMs to refuse to answer questions where it has a low confidence. There are two key underlying ideas. Zhang et al. (2024a), Cohen et al. (2024), Kapoor et al. (2025) identify such questions according to answer correctness and pad sure or unsure to the end of the answer. Cheng et al. (2024); Yang et al. (2024b) in addition require providing the correct answer consistently. Our CONFQA training is tremendously different in two ways: first, we use the *dampener* prompt, which reduces hallucination further by 5-11% in our empirical study; second, we focus on simple factual questions from the DBPedia knowledge graph, which increases factuality by up to 30%. Our experiments also show that requiring consistency in addition can cause a large correctness regression. We present the experimental comparison in detail in Section 5.

**LLM hallucination suppression:** Training-based hallucination mitigation has been surveyed in Tonmoy et al. (2024). There are two directions for training: *enriching the parameterized knowledge*, and *suppressing hallucinations*, the latter more related to our work. In addition to the aforementioned methods that teach LLMs about its confidence, Sun et al. (2023b) teaches LLMs to *recite* factual passages to avoid hallucination. Dhuliawala et al. (2023) verifies responses with internalized knowledge before final generation. Tian et al. (2023a) generates factuality preference rankings to favor factual statements consistent with external sources or internal knowledge. Xie et al. (2025) trains a factuality evaluator to provide LLM generators with claim-level factuality feedback. Grattafiori et al. (2024) incorporates refusals in training data for samples that got consistently informative and incorrect responses, similar as in Cheng et al. (2024).

## 3 METHODOLOGY

### 3.1 PROBLEM DEFINITION

Consider a *Factual Question Answering (QA)* problem: given a question $Q$ that asks for factual information like the director of a movie, generate an answer $A$ with precise information. An LLM-based system can answer the question based on its internal knowledge: $A = M(Q)$, where $M$ denotes the model; or take the RAG approach and resort to external information repository $\mathcal{R}$: $A = RAG(Q, \mathcal{R})$. For simplicity we assume optimal retrieval and augmentation in the RAG pipeline. Whereas the RAG approach yields higher-quality answers, it also incurs large overheads such as retrieval latency; therefore, we shall trigger RAG only when necessary.

The *RAG Triggering* problem takes an input question $Q$ and outputs a boolean regarding whether to trigger RAG: $T : Q \rightarrow \{0, 1\}$. In other words,

$$A = \begin{cases} RAG(Q, \mathcal{R}), & \text{if } T(Q) = 1, \\ M(Q), & \text{if } T(Q) = 0. \end{cases} \quad (1)$$

An ideal triggering strategy should invoke RAG only when the LLM's inherit answer $M(Q)$ is incorrect, thereby minimizing unnecessary overhead while optimizing quality. This reduces the probelm to train the LLM $M$ to recognize when its own output may be unreliable and to return an *"Unsure"* response in such cases. This response then serves as a signal to trigger RAG. We next describe CONFQA, a fine-tuning approach for calibrating model confidence, and describe how it supports our triggering strategy.

### 3.2 CONFQA: WHEN TO SAY UNSURE?

The goal of CONFQA is to fine tune an LLM to only answer a question that it has high confidence about, and admitting *"I am unsure"* otherwise. We have three key intuitions for this fine-tuning. First, we calibrate the LLM's confidence by showing the ground truth. Second, we introduce a *dampener prompt "Answer only if you are confident"*, to explicitly guide LLM's behavior. Third, as we wish to regularize behavior only for factual statements, we focus the teaching on atomic facts (attributes of entities) to avoid distractions of other factors.

We prepare the training data as a collection of question–label pairs, where each label provides the model with the appropriate response to generate. The questions ask for atomic facts, and are generated from DBPedia, which covers a diverse set of domains (Intuition #3). We used the open-sourced script from Sun et al. (2023a) to generate question-answer pairs from DBPedia, evenly distributed among different entity popularity: head, torso, and tail entities.

We generate labels as follows. First, we prompt Llama-3.1-70B model to answer the DBPedia-based questions (Prompt 1 in Appendix A). Then, we prompt Llama-3.1-405B to judge if the answer is consistent with the ground truth (Prompt 2 in Appendix A). If the answer is correct, the label is the ground truth answer; otherwise, the label is *"I am unsure about the answer"* (Intuition #1).

We provide the dampener prompt as the system prompt both in training and in inference, as an explicit instruction for the model to suppress hallucinations (Prompt 1 in Appendix A). We call our fine-tuned model CONFQA, denoted by $\hat{M}$.

### 3.3 CONFRAG: WHEN TO TRIGGER RAG?

Since CONFQA is fine-tuned to answer a question only if it is confident, we can invoke the RAG pipeline when it says unsure, and rely on the LLM-generated answer otherwise.

$$A = \begin{cases} RAG(Q, \mathcal{R}), & \text{if } \hat{M}(Q) = \text{"unsure"}, \\ \hat{M}(Q), & \text{otherwise.} \end{cases} \quad (2)$$

Figure 2 depicts The CONFRAG invocation architecture. For *dynamic* questions—those requiring up-to-date information—the system always responds through the RAG pipeline. For *static* questions, the system runs LLM generation ($\hat{M}$) and the RAG pipeline in parallel. If the LLM produces

| Benchmark | Category | Question types | # Domain | Size |
|---|---|---|---|---|
| Head-to-Tail (Sun et al., 2023a) | short-form | simple questions (attribute of entities) | dbpedia, imdb | 1,200 |
| SimpleQA (Wei et al., 2024a) | short-form | general fact-seeking questions | multiple domains | 4,326 |
| CRAG (Yang et al., 2024a) | short-form | simple questions, reasoning questions | 5 domains | 642 |
| LongFact (Wei et al., 2024b) | long-form | general questions | 38 domains | 250 |
| AlpacaFact (Lin et al., 2024) | long-form | fack-seeking instruction-following | multiple domains | 241 |
| Biography (Min et al., 2023) | long-form | biography questions | celebrity | 183 |
| MMLU 5-shot (Hendrycks et al., 2021) | general knowl. | multi-choice questions | 57 domains | 14,042 |
| MMLU pro (Wang et al., 2024) | general knowl. | multi-choice questions | multiple domains | 12,032 |

Table 1: The overall statistics of evaluation datasets.

a valid answer, the RAG process is early-stopped and the LLM output is returned; otherwise, the system waits for and outputs the RAG result.

# 4 EXPERIMENT SETUP

## 4.1 BENCHMARKS AND METRICS

**Data sets:** We experimented with three *Short-form factuality benchmarks*, where the answers are mostly short; the question include both *simple* questions regarding an attribute of an entity, and *complex* ones that require comparison, aggregation, reasoning, and post-processing. Table 1 summarizes the benchmarks and Appendix B gives details.

**Metrics and evaluation:** For model metrics, following the CRAG benchmark (Yang et al., 2024a), we compute the percentage of *correct*, *incorrect* (i.e., hallucinations), and *missing* (not attempted) answers, and take *Factuality* = correct% - incorrect% as our major metrics. Factuality ranges from -1 to 1 and penalizes hallucinations more than missing answers. We use prompt based LLM-as-a-judge to evaluate model answers. As observed in Yang et al. (2024a), LLM-as-a-judge achieves 99% accuracy.

For triggering, we compute the precision and recall of the triggering decisions compared to the oracle solution that triggers when an answer is incorrect or missing. Take the DBPedia results in Table 2 as an example. The ground truth triggering is 48.0 (sum of Miss and Incor of Llama-3.1).

*Precision* computes how many oracle triggers are indeed triggered, whereas Recall computes how many triggers are needed. *F-measure* computes their harmonic mean: $F_{msr} = \frac{2 \cdot prec \cdot rec}{prec + rec}$. Take the CONFQA model results for DBPedia in Table 2, CONFRAG triggers 63.3 percentage, and Incorrect is 5.2. Thus *Precision* equals $\frac{min\{63.3, 48\}}{63.3} = 75.8\%$, *Recall* equals $\frac{min\{63.3, 48\}}{min\{63.3, 48\} + 5.2} = 90.2\%$ and $F_{msr} = 82.4\%$. See Section E for detailed explanation.

**LLM Models and implementations:** We conduct experiments using six well-known LLMs: Llama3.1-8B, Llama3.1-70B (Touvron et al., 2023), GPT-4o-mini and GPT-4o (OpenAI et al., 2024), Claude3.5-Sonnet[1] and Claude3.5-Haiku[2].

Our fine-tuning uses Llama-3.1-70B as the backbone[3], and have observed similar results when fine-tuning Llama-3.1-8B. We conducted a simple scaling-law study (see Appendix F) and decided to run one epoch on 3K samples of training data, with a learning rate of 1e-6, a batch size of 1, a gradient accumulation step 1. In addition to Llama models, we also fine tuned QWen2.5-7B-Instruct and Gemma-3-4B-IT. Experiment results are discussed in Appendix I.

We conducted experiments on Nvidia H100 96GB HBM2e GPUs with different configurations. For Llama3.1-70B models, we fine tuned on 32 GPUs and inference on 8 GPUs.

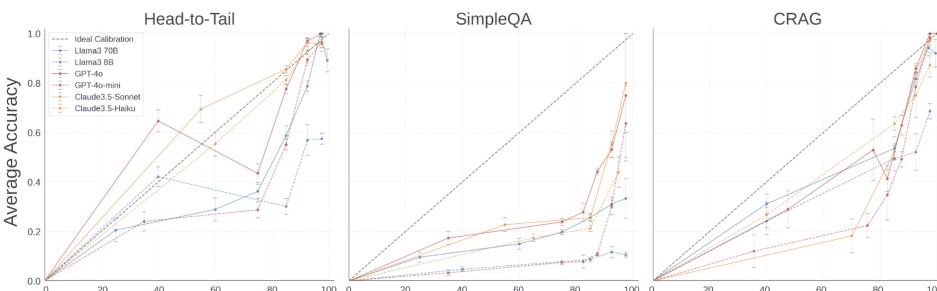

Figure 3: LLMs' self-reported confidences is correlated with QA accuracy, but often over-confident.

## 4.2 ANALYSIS OF LLM'S SELF-REPORTED CONFIDENCE (RQ1)

With the three short-form benchmarks, we answer the first research questions:

- **RQ1**. *Does an LLM know what it knows?*

We prompted the LLM to directly provide a confidence score between 0 and 1 along with its answer (prompt template in Appendix 3). We removed missing answers, divide the reported confidences by equal-sized quantile bins, and plot the average accuracy within each bin. We investigated the *calibration* between confidence and QA accuracy; that is, when the model has a confidence of 0.8, is the QA accuracy close to 80%? Figure 3 shows the calibration, leading to three observations. (We in addition compare calibration for head, torso, tail entities in Appendix C and with consistency of answers in Appendix D).

1. The self-reported confidence is mostly positively correlated with QA accuracy, but LLMs tend to be over-confident (the correlation curves are below the ideal calibration dashed line). For example, when Llama-3.1-70B predicts a confidence of 80% on CRAG, the real accuracy is only 33%.

2. Notably, for the same model series, the smaller model is often more confident than the larger model (with an exception of Claude3.5 on CRAG), demonstrating the interesting correlation between ignorance and self-assurance.

3. Finally, the overconfidence is more pronounced when answering SimpleQA questions than on other benchmarks. A sample of 50 questions from SimpleQA shows that the questions are often nuanced for fairly popular entities (e.g. *"What was the first line after the salutation in the letter sent to Sardar Patel by Abhay Charan De?", "In which month and year was Service Pack 3 for Windows Fundamentals for Legacy PCs released?"*), possibly causing LLMs to be over-confident.

As this analysis shows, *self-reported confidence* tends to be over-confident so cannot serve the purpose of RAG triggering decision, justifying the need for fine-tuning.

## 5 EXPERIMENTAL RESULTS

In this section, we study the effectiveness of CONFQA and CONFRAG, and answer the following two research questions.

- **RQ2**. *Can we teach LLMs to refrain from hallucinations?*

- **RQ3**. *What is the optimal strategy for RAG triggering?*

---

[1] https://www.anthropic.com/news/claude-3-5-sonnet

[2] https://www.anthropic.com/claude/haiku

[3] we also fine tuned Gemma and QWen, results will be shared upon request.

| Model | Corr | Miss | Incor | Fac. | Ceiling Corr | Ceiling Fac. | Tri. Prec | Tri. Rec | $F_{msr}$ |
|---|---|---|---|---|---|---|---|---|---|
| | | | | | **DBpedia** (in-domain) | | | | |
| Llama-3.1 | **52.0** | **22.0** | 26.0 | 26.0 | 74.0 | 48.0 | - | - | - |
| Llama-3.1 Dampen | 47.0 | 26.8 | 26.2 | 20.8 | 73.8 | 47.6 | **100.0** | 50.6 | 67.2 |
| R-tuning (MMLU) | 24.3 | 67.3 | 8.3 | 16.0 | 91.6 | 83.3 | 71.3 | 85.3 | 77.7 |
| R-tuning (DBPedia) | 24.5 | 67.8 | 7.7 | 16.8 | 92.3 | 84.6 | 70.8 | 86.2 | 77.7 |
| IDK (DBPedia) | 17.0 | 81.5 | **1.5** | 15.5 | 98.5 | 97.0 | 58.9 | **97.0** | 73.3 |
| CONFQA | 31.5 | 63.3 | _5.2_ | **26.3** | 94.8 | 89.6 | 75.8 | 90.2 | **82.4** |
| | | | | | **IMDB** (out-of-domain) | | | | |
| Llama-3.1 | **44.8** | **34.2** | 21.0 | 23.8 | 79.0 | 58.0 | - | - | - |
| Llama-3.1 Dampen | 40.7 | 36.2 | 23.2 | 17.5 | 76.9 | 53.7 | **100.0** | 60.9 | 75.7 |
| R-tuning (MMLU) | 28.2 | 60.5 | 11.3 | 16.9 | 88.7 | 77.4 | 91.2 | 83.0 | 86.9 |
| R-tuning (DBPedia) | 25.3 | 70.2 | 4.5 | 20.8 | 95.5 | 91.0 | 78.6 | 92.5 | 85.0 |
| IDK (DBPedia) | 22.0 | 77.0 | **1.0** | 21.0 | 99.0 | 98.0 | 71.7 | **98.2** | 82.9 |
| CONFQA | 32.5 | 63.3 | _4.2_ | **28.3** | 95.8 | 91.6 | 87.2 | 92.9 | **90.0** |
| | | | | | **SimpleQA** (out-of-domain) | | | | |
| Llama3.1 | 20.0 | 44.1 | 35.9 | -15.8 | 64.1 | 28.2 | - | - | - |
| Llama3.1 Dampen | 16.8 | 48.0 | 35.2 | -18.4 | 64.8 | 29.6 | **100.0** | 57.7 | 73.2 |
| R-tuning (MMLU) | **20.3** | **38.0** | 41.7 | -21.4 | 58.3 | 16.6 | **100.0** | 47.7 | 64.6 |
| R-tuning (DBPedia) | 3.7 | 83.3 | 13.0 | -9.3 | 87.0 | 74.0 | 96.0 | 86.0 | 90.8 |
| IDK (DBPedia) | 0.6 | 99.1 | **0.2** | 0.4 | 99.7 | 99.5 | 80.7 | **99.8** | 89.2 |
| CONFQA | 4.9 | 93.1 | _2.1_ | **2.8** | 98.0 | 95.9 | 85.9 | 97.4 | **91.3** |
| | | | | | **CRAG** (out-of-domain) | | | | |
| Llama3.1 | **58.7** | 15.6 | 25.7 | 33.0 | 74.3 | 48.6 | - | - | - |
| Llama3.1 Dampen | 57.5 | 22.3 | 20.2 | **37.2** | 79.8 | 59.6 | **100.0** | 52.5 | 68.8 |
| R-tuning (MMLU) | 57.8 | **17.1** | 25.1 | 32.7 | 74.9 | 49.8 | **100.0** | 40.5 | 57.7 |
| R-tuning (DBPedia) | 31.6 | 55.0 | 13.4 | 18.2 | 86.6 | 60.9 | 75.1 | 75.5 | 75.3 |
| IDK (DBPedia) | 20.7 | 78.2 | **1.1** | 19.6 | 98.9 | 97.8 | 52.8 | **97.4** | 68.5 |
| CONFQA | 39.4 | 56.2 | _4.4_ | 35.0 | 95.6 | 91.2 | 73.5 | 90.4 | **81.1** |

Table 2: Overall factuality and triggering $F_{msr}$ improvement on short-form benchmarks; CONFQA can reduce hallucination to below 5%. CONFRAG achieves the best $F_{msr}$ over all methods among all benchmarks. The optimal $F_{msr}$, Tri. Prec, Tri. Rec, Factuality and Incorrect rate are shown in bold. The second best Incorrect rate are shown in italic. Dash - indicates the metrics are not valid as as we use Llama-3.1 triggering as ground truth. All numbers are in percentage (%).

## 5.1 ALTERNATIVE SOLUTIONS

We compare CONFQA with two baseline solutions: LLM without dampener and LLM with dampener in the inference. In addition, we implemented two state-of-the-art solutions. **R-Tuning** (Zhang et al., 2024a) generates its training data by adding a prompt "Are you sure you accurately answered the question based on your internal knowledge?" in the question, and padding "I am sure" or "I am unsure" based on correctness of the generated answer. In the inference, it again appends the prompt and applies post-processing by removing answers with the suffix of "I am unsure". We used both MMLU (Hendrycks et al., 2021) (proposed in the paper) and DBPedia for training. **IDK** (Cheng et al., 2024) requires answer consistency in addition to answer correctness, and we add a consistency requirement of at least four out of five times. We used DBPedia for its fine-tuning for more direct comparison.

## 5.2 EFFECTIVENESS OF CONFQA (RQ2)

Table 2 left part presents the answer quality from CONFQA. First, baseline methods are hallucination prone. Without fine-tuning, the effect of the dampener is inconsistent. For all benchmarks, the dampener increased the percentage of missing answers by 2-7%. However, it (correctly) dampens hallucinations on CRAG, but dampens correct answers and thus reduced the factuality on Head-to-tail (DBPedia) and SimpleQA. This is not surprising since the LLM confidence is not well calibrated.

| Model | SimpleQA | | | | | | | CRAG | | | | | | |
|---|---|---|---|---|---|---|---|---|---|---|---|---|---|---|
| | Upper | Corr | Miss | Incor | Fac | L-P50 | L-P90 | Upper | Corr | Miss | Incor | Fac | L-P50 | L-P90 |
| LLM-only | 20.0 | 20.0 | 44.1 | 35.9 | -15.8 | 480 | 896 | 58.7 | 58.7 | 15.6 | 25.7 | 33.0 | 480 | 896 |
| RAG-everywhere | **100.0** | **78.1** | 11.5 | **10.5** | **67.6** | 1,900 | 2,780 | **100.0** | 61.1 | 15.1 | 23.8 | 37.3 | 1,900 | 2,780 |
| CONFQA-based | 95.1 | 77.2 | **11.4** | 11.5 | 65.7 | 1,802 | 2,650 | 95.6 | **62.3** | **14.2** | *23.5* | **38.8** | 1,278 | 1,955 |

Table 3: CONFQA-based RAG invocation achieves similar quality to RAG-everywhere, while cutting latency. **Upper**: upper bound of percentage of correct. **L-P50**: P50 latency and **L-P90**: P90 latency.

| Model | DBpedia (in-domain) | | | | IMDB (out-of-domain) | | | | SimpleQA (out-of-domain) | | | | CRAG (out-of-domain) | | | |
|---|---|---|---|---|---|---|---|---|---|---|---|---|---|---|---|---|
| | Corr | Miss | Incor | Fac | Corr | Miss | Incor | Fac | Corr | Miss | Incor | Fac | Corr | Miss | Incor | Fac |
| Llama-3.1 Dampen | 47.0 | 26.8 | 26.2 | 20.8 | 40.7 | 36.2 | 23.2 | 17.5 | 16.8 | 48.0 | 35.2 | -18.4 | **57.5** | 22.3 | 20.2 | 37.2 |
| CONFQA | 31.5 | 63.3 | **5.2** | 26.3 | 32.5 | 63.3 | **4.2** | **28.3** | 4.9 | 93.1 | **2.1** | **2.8** | 39.4 | 56.2 | **4.4** | 35.0 |
| * No-dampener in Inf. | **49.9** | 33.5 | 17.5 | **31.5** | **43.0** | 42.0 | 16.0 | 27.0 | 17.3 | 55.8 | 26.8 | -9.5 | 57.0 | 19.6 | 23.4 | 33.6 |
| * No-dampener in Train | 36.0 | 50.2 | 13.8 | 22.2 | 34.2 | 50.7 | 15.2 | 19.0 | 5.8 | 87.5 | 6.7 | -0.9 | 46.0 | 44.2 | 9.8 | **36.2** |
| * MMLU-as-source | 8.2 | 89.8 | **2.0** | 6.2 | 16.7 | 82.0 | **1.3** | 15.4 | 0.6 | 98.8 | **0.5** | 0.1 | 7.0 | 92.7 | **0.3** | 6.7 |
| * GT-as-label | 48.0 | 2.8 | 49.2 | -1.2 | 41.2 | 4.3 | 54.5 | -13.3 | **17.8** | 13.7 | 68.4 | -50.6 | 53.7 | 14.2 | 32.1 | 21.6 |

Table 4: Ablation study for CONFQA when applying dampener in inference for all models (unless explicitly says no), showing effectiveness of our fine-tuned model. Optimal Correct, Factuality and two best Incorrect are shown in bold. All numbers are in percentage (%) and full results in Table 8.

Second, the fine-tuned CONFQA improves factuality by up to 20%, and the hallucination rate drops to below 5% on all benchmarks. As a side effect, correctness also drops; for example, since SimpleQA focuses on nuanced facts, after the finetuning we observe nearly zero correctness. Still, there is much more dropping on hallucniated answers than on correct answers, and thus the factuality increases across benchmarks.

Third, R-tuning mostly has lower hallucination, especially if trained on DBPedia. However, we also observe much lower correctness. We suspect this is because when the model gives a wrong answer, the training data feed ground truths as additional knowledge and causes confusion. We also observe stronger performance when trained on DBPedia than on MMLU, as MMLU mixes facts with reasoning skills and can introduce ambiguity, supporting our hypothesis that atomic facts yield better training examples.

Fourth, IDK obtains the lowest hallucination rate (below 1.5% for all benchmarks), as it requires in addition the consistency signal and thus is stricter. However, the correctness also drops significantly, reducing overall factuality.

Finally, even though the training data are generated only from DBPedia, CONFQA behavior changes on the other datasets as well, showing amazing generalization.

## 5.3 EFFECTIVENESS OF CONFRAG (RQ3)

Table 2 also compares the effectiveness of triggering using different QA solutions, showing that CONFRAG has the highest triggering F-measure for each datasets, enables potential truthfulness gains to beyond 95%, while reducing unnecessary external retrievals by 5-19%. We have also compared CONFRAG with the prompt based triggering methods in Ni et al. (2024), and results are shown in Appendix Table 12.

We next evaluate QA quality and latency through a real RAG implementation, which invokes search APIs (Bing API and Knowledge Graph API) for retrieval, and passes the retrieved content to Llama-3.1-70B to generate the responses. Table 3 reports the end-to-end QA accuracy and latency for our proposed RAG architecture, and compares it with not invoking RAG and invoking RAG everywhere. CONFQA-triggered RAG obtains similar quality to triggering RAG everywhere, but reduced latency by 600ms P50 and 800ms P90 for CRAG. The SimpleQA benchmark requires triggering RAG for the majority of the questions to achieve high quality; the latency improvement is less pronounced but our triggering does not regress quality compared to RAG-everywhere. The CRAG benchmark contains a lot of complex questions requiring reasoning over retrieval results; our simple RAG implementation does not excel, but still improves on factuality.

| (a) DBPedia | (b) IMDB | (c) CRAG |

Figure 4: CONFQA suppresses more (gives more missing answers) on less popular entities (missing rate for Tail is larger than Torso, whose missing rate is larger than Head).

| Model | Long Fact | | | | Alpaca Fact | | | | Biography | | | |
|---|---|---|---|---|---|---|---|---|---|---|---|---|
| | Prec | Rec | F1 | Miss | Prec | Rec | F1 | Miss | Prec | Rec | F1 | Miss |
| Llama3.1 | 64.5 | 65.4 | 64.3 | 0 | 62.3 | 71.0 | 63.8 | 0 | 35.4 | 40.3 | 37.1 | 0 |
| RAG (Llama3.1) (Yu et al., 2022) | 71.7 | 74.6 | 72.7 | 0 | 65.8 | 74.3 | 66.0 | 0 | 44.9 | 48.1 | 43.8 | 0 |
| CONFQA | 67.0 | 67.7 | 66.7 | 0.8 | 62.2 | 71.1 | 63.8 | 0.4 | 42.0 | 46.5 | 42.6 | 12.6 |

Table 5: CONFQA improves precision and recall for long-form answer generation.

## 5.4 DEEP DIVE ON CONFQA

**Ablation study** We now compare CONFQA with the several alternatives, as shown in Table 4 (full comparison in Table 8 in Appendix H). First, without the dampener in inference, we observe minor sacrifice or even slight increase on correct answers, but also just mild reduction of the hallucinations. On the other hand, without the dampener in training, we observe increased correctness and reduced missing rate, but also increased hallucinations than CONFQA and thus lower factuality, showing the important role of the dampener in training as well.

Use MMLU, instead of DBPedia, to generate training data obtains low hallucination (below 2%) but also significantly lower correctness. We suspect this is because MMLU contains a diverse set of tasks, reducing overall confidence of the model. *GT-as-label* achieves high correctness and lowest missing rate, but becomes over-confident to hallucinate (hallucination rate can reach 70%). This is consistent with observations in previous work (Lin et al., 2024; Gekhman et al., 2024) that feeding facts in the SFT-stage can teach LLMs to hallucinate.

**Answer distributions:** Finally, we show in Figure 4 the distribution of correct, missing, and incorrect answers for entities of different popularity (Head, Torso, Tail), before and after fine-tuning, with and without dampening. It confirms that fine-tuning suppresses hallucinations, and the dampener prompt further reduces hallucinations. Additionally, it shows CONFQA suppresses more on long-tail facts, where it lacks confidence.

## 6 DISCUSSIONS

In addition to triggering, we also examined CONFQA on other benchmarks, to understand whether it can apply to suppress hallucination when access to retrieval corpus is unavailable, such as in most speech-in speech-out systems (Xie & Wu, 2024; Nguyen et al., 2022; Zhang et al., 2024b). We wish to reduce hallucinations on factual statements, without regressing performance on general knowledge and problem-solving tasks.

We thus consider benchmarks in two other categories: 1) *Long-form factuality benchmarks*, where answers are expected to be long and contain multiple factual statements. We use the automatic evaluation metric, *VeriScore* (Song et al., 2024), which computes precision, recall, and F1-score. We set the minimum number of facts required for perfect recall based on the median number of extracted claims per dataset,

| Model | MMLU (5-shot) | MMLU-Pro |
|---|---|---|
| Llama3.1 | 82.7 | 66.3 |
| CONFQA | 82.8 | 65.4 |

Table 6: CONFQA does not regress on MMLU.

using their fine-tuned models for claim extraction and verification. 2) *General knowledge benchmarks*, which focuses on general knowledge and reasoning in diverse disciplines. MMLU provides ground truths for the multi-choice questions. The score is computed as the percentage of correctly answered questions, as a weighted average among the 57 diverse subjects. Again, Table 1 summarizes the benchmarks and Appendix B gives details.

On the long-form benchmarks, we do not apply the dampener prompt, and instead we retrieve 10 passages using the input prompts as queries and append *"Provide as many specific details and examples as possible (such as names of people, numbers, events, locations, dates, times, etc.)"* to the end of the prompt to encourage the model to provide as much confident information as possible. Table 5 shows that CONFQA achieves higher or comparable precision and recall, except for 13% biography questions where it has low confidence about and does not answer. For RAG, we use Contriever (Izacard et al., 2022) to retrieve passages from C4 (Raffel et al., 2020) and Wikipedia, following the setting in MassiveDS (Shao et al., 2024).

We also evaluate CONFQA on the standard MMLU benchmark, and also do not apply the dampener prompt. Table 6 shows that the scores are mostly similar to the baseline.

Together with the results presented in Section 5, the experiments suggest another potential application of CONFQA when RAG is not an option. If one aims to maintain the correctness and meanwhile reduce hallucinations as much as possible, we recommend CONFQA without the dampener; if one emphasizes hallucination elimination, we recommend applying CONFQA with the dampener.

## 7  LIMITATIONS

Our experiments focus on SFT, and we leave DPO-based fine-tuning for future work. We empirically compared DBPedia and MMLU, where the former contains only simple factual questions, and the latter contains questions ranging from factual to reasoning. A comprehensive study regarding the effect of sources with different coverage in this spectrum would improve the understanding. We can extend our learning beyond factual statements, for math, coding, reasoning etc. Lastly, SFT requires the access to LLM itself for fine tuning, and thus restrict the application of the proposed framework to proprietary LLMs only accessible through APIs.

## 8  CONCLUSION

Recent studies have shown that LLMs acquire substantial knowledge during pre-training, and that introducing new knowledge during post-training can often increase hallucinations (Lin et al., 2024; Gekhman et al., 2024). Rather than injecting additional knowledge, the CONFRAG framework we propose trains LLMs to identify and withhold low-confidence factual claims, deferring to RAG in such cases to improve factuality. Through a comprehensive set of experiments, we show that 1) LLMs tend to be over-confident on what they know; 2) we can fine-tune LLM to refrain from generating inconfident factual statements and thus reducing the hallucination rate to below 5%; and 3) using this fine-tuned model for RAG-triggering can reach similar answer accuracy as RAG-everywhere, whereas reducing RAG retrievals by over 5-19% to save cost. Our proposed framework naturally integrates internal neural knowledge with external symbolic knowledge, allowing for improvements in both factuality and latency as LLMs develop richer internal knowledge and more effective RAG capabilities.

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

# A PROMPTS

We have included a list of prompts used in our experiments for reference. Each prompt is cited in the main paper where it is applied. For clarity, we also provide brief descriptions of each prompt here.

### Prompt 1: Simple question answer generation prompt

---
As Assistant AI, you help answer factual questions. Please keep your responses short and concise and directly provide the answer to the user question without reasoning. Answer only if you are confident; otherwise, respond with 'I am unsure about the answer'.

---

Prompt 1 has been used in multiple places.

- When generating training data, we use this prompt to prompt Llama-3.1-70B to answer the 3k DBPedia-based questions.
- During SFT, we pass in this prompt as system prompt, unless we explicitly say no prompt used in training.
- During benchmarks, we pass this as system prompt in inference, unless we explicitly say no prompt used.

Prompt 2: Simple question answer grading prompt. Used in Section 3.2 to generate training labels: grade if the model answer is consistent with the ground truth. If yes, use ground truth as the label, otherwise, use *"I am unsure about the answer"* as the label.

---
You need to check whether the prediction of a question-answering system to a question is correct. You should make the judgment based on a list of ground truth answers provided to you. Your response should be "correct" if the prediction is correct or "incorrect" if the prediction is wrong.

Example 1:
Question: Who authored The Taming of the Shrew (published in 2002)? Ground truth: ["William Shakespeare", "Roma Gill"]
Prediction: W Shakespeare
Correctness: correct

Example 2:
Question: Who authored The Taming of the Shrew (published in 2002)? Ground truth: ["William Shakespeare", "Roma Gill"]
Prediction: Roma Gill and W Shakespeare
Correctness: correct

Example 3:
Question: Who authored The Taming of the Shrew (published in 2002)? Ground truth: ["William Shakespeare", "Roma Gill"]
Prediction: Roma Shakespeare
Correctness: incorrect

Example 4:
Question: What country is Maharashtra Metro Rail Corporation Limited located in?
Ground truth: ["India"]
Prediction: Maharashtra
Correctness: incorrect

Example 5:
Question: What's the job of Song Kang-ho in Parasite (2019)? Ground truth: ["actor"]
Prediction: He plays the role of Kim Ki-taek, the patriarch of the Kim family.
 Correctness: correct

---

Prompt 2 has been used in two places:

- When generating training data, this prompt is used to grade if model answer is the same as ground truth.
- During benchmarks of evaluating different models , this is used to grade the model responses.

Prompt 3 is used in Section 4.2 to prompt the model to output confidence score, together with an answer. This is used to study the relationship of model self-reported confidence with answer accuracy. Same prompt is also used to study the relationship of consistency of model answers with answer accuracy in Section D.

Prompt 3: Self-reported confidence prompt.

---

Please answer the following question truthfully to your best knowledge. Provide a confidence score between 0 to 1 representing your confidence in the answer.
Question: {{{question}}}
Your response must be in the following JSON format.
{"answer": "Your answer here", "confidence_score": float number}
Remember, your answer MUST be a valid json string with two keys: "answer", "confidence_score".

---

## B  BENCHMARK SETUP DETAILS

To evaluate the performance of the fine tuning, We consider 7 benchmarks in the main content as described in Section 4.1, with details described here.

- **Head-to-Tail (*short-form* with simple questions)** (Sun et al., 2023a): We leveraged the data scraping scripts from the HEAD-TO-TAIL benchmark, and generated simple questions and their answers according to content from DBPedia [4] (general knowledge graph) and IMDb [5] (data in the *Movie* domain). From each dataset we randomly sampled 200 entities for *head* entities, 200 for *torso* entities, and 200 for *tail* entities. Here we follow the definition in Sun et al. (2023a) for head, torso and tail: we rank all entities by their traffic; head entities are top-popular entities that together account for 1/3 of traffic, tail entities are unpopular entities that together account for 1/3 of traffic, and torso entities are the remaining medium-popular entities. Together, we have 1200 question-answer pairs, 600 from each source.

- **SimpleQA (*short-form* with simple questions)** (Wei et al., 2024a): SIMPLEQA is a benchmark released by *OpenAI* to measure LLM factuality. It contains 4326 manually crafted short, fact-seeking questions, covering diverse topics such as science, technology, history, and entertainment.

- **CRAG (*short-form* with simple and complex questions)** (Yang et al., 2024a): CRAG is a benchmark to test RAG capabilities. It contains 4,409 training and 1335 evaluation questions covering five domains (general, finance, sports, music, movie), entities of different popularities (head, torso, tail), facts of different dynamisms (static, slow-changing, fast-changing, real-time), and eight question types (simple, condition, set, comparison, aggregation, multi-hop, post-processing, false premise). We selected the 642 *static* questions from the evaluation data set, with 97 questions for head entities, 99 for torso, 90 for tail entities and 356 for facts from the web (mostly popular); we excluded false-premise and dynamic questions from the sampling as it presents different challenges.

- **LongFact (*long-form*)** (Wei et al., 2024b): Aiming to measure of the factuality of long-form responses consisting of at least several paragraphs, LongFact has 2,280 factual questions covering 38 topics, generated by prompting GPT-4. Following Wei et al. (2024b), we use the 250 prompts from the LongFact-Objects dataset in our experiments.

- **AlpacaFact (*long-form*)** (Lin et al., 2024): Initially sourced from diverse interactions with real-world users, the 805 instructions in AlpacaFarm (Dubois et al., 2023) served as a benchmark for evaluating the ability of different LLMs to follow instructions. Following Lin et al. (2024), we used a subset of 241 fact-seeking instructions in this work.

- **Biography (*long-form*)** (Min et al., 2023): To validate the effectiveness of FActScore, Min et al. (2023) created a collection of prompts named Biography by applying the template "`Tell me a bio of [Person Name]`" to 183 notable individuals listed on Wikipedia. Given its extensive use in recent literature, we have included this prompt set for our experiments as well.

- **MMLU (*General knowledge*)**: The MMLU (Hendrycks et al., 2021) dataset covers 57 subjects, including areas such as mathematics, history, law, and medicine. It contains two subsets: the MMLU 5-shots dataset contains 14,042 multi-choice questions to evaluate general knowledge and problem-solving tasks; the MMLU-Pro (Wang et al., 2024) dataset

---

[4] `dbpedia.org`
[5] `imdb.com`

contains 12,082 multi-choice questions to stress-test reasoning, disambiguation, and factual accuracy.

For short-form questions we consider factuality score as defined in Section 4.1 rather than F1 score, where F1-score is more lenient for incorrect answers (hallucinations), but factuality strongly prefers missing answers to hallucinations. For example, consider a model that answers 10% questions correctly (correct% = 10%) and the rest of the questions incorrectly (incorrect% = 90%); the F1-score is 10% (not punishing hallucinations much) while the factuality is -80%. Now consider another models that answers 10% questions correctly and admits "I an unsure about the answer" for the rest of the questions; the F1-score is 18.2%, only slightly higher than 10%, but the factuality is 10%, significantly higher than -80%.

For long-form responses we use the automatic evaluation metric, *VeriScore* Song et al. (2024), for measuring the factuality. Following FActScore (Min et al., 2023) and SAFE (Wei et al., 2024b), VeriScore extracts more sensible and verifiable claims from each sentence and uses Google search snippet instead of Wikipedia as the source of knowledge. This approach allows VeriScore to be applied to more diverse topics and requires fewer but more meaningful claims to be checked. We report the F1 score from VeriScore, which represents the harmonic mean of precision and recall. In line with Song et al. (2024), we set the minimum number of facts required for perfect recall based on the median number of extracted claims per dataset, using their fine-tuned models for claim extraction and verification.

## C INFLUENCE OF ENTITY POPULARITY ON CONFIDENCE

In this section, we study the calibration versus popularity of the entities. Figure 5 show the calibration on the Head-to-Tail and CRAG benchmarks, where questions are categorized by entity popularity into *Head, Torso, Tail* (plus *Web* for CRAG).

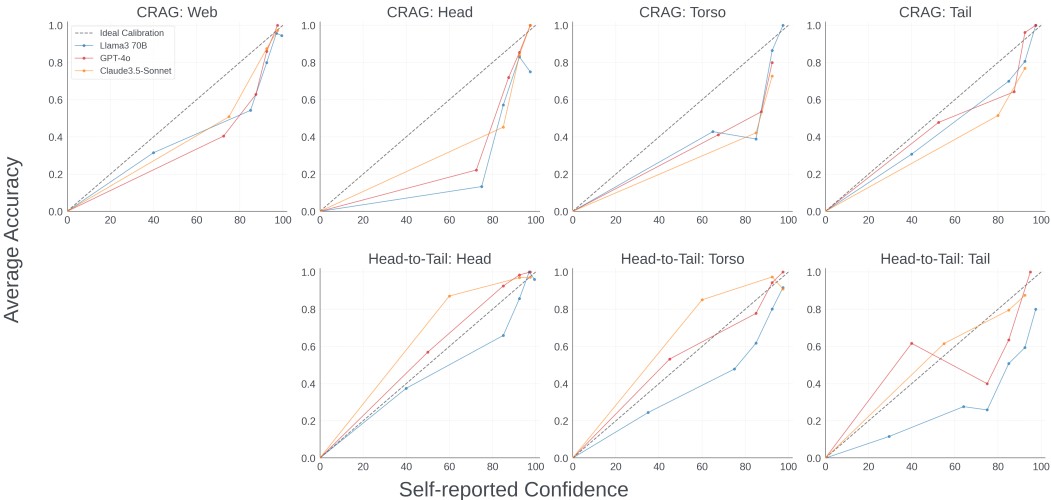

Figure 5: Correlation between LLM's self-reported confidences and average accuracies on the CRAG dataset and the Head-to-Tail dataset, categorized by question types.

Interestingly, we found for simple questions on Head-to-Tail, models are better calibrated for head entities than torso or tail entities (Figure 5 bottom panels). However, on more complex questions on CRAG, models are better calibrated for tail entities than torso or head entities (Figure 5 top panels). This shows two different dimensions that can affect the model confidence: entity popularity and question nuances.

## D    INFLUENCE OF ANSWER CONSISTENCY ON CONFIDENCE

**Consistency vs. Accuracy:**    In this section, we study the LLM's answers consistency versus cali-
bration. To measure consistency, we ask LLM the same question 20 times with the temperature set
to 1.0, select the most frequent response as the final answer, and calculate its frequency among the
20 times as the consistency score. To be robust against minor differences, we determine the "most
frequent" answer based on semantic similarity rather than exact string match.

Figure 6 shows that consistency is mostly better calibrated than self-reported confidence, and largely
overlays with the ideal calibration on CRAG. In addition, the calibration curve is more linear com-
pared to self-reported confidence in Figure 3.

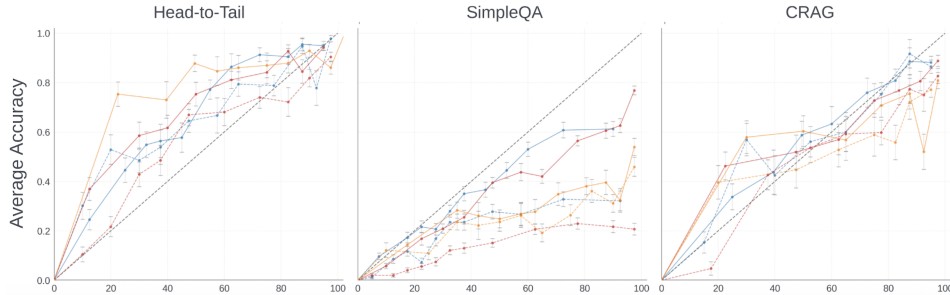

Figure 6: LLM's answers consistency is often better calibrated with QA accuracy than self-reported
confidences.

## E    TRIGGERING METRICS DEFINITION

In Section 4.1, we define the Triggering *Precision*, *Recall* and $F_{msr}$. We will explain more intuitively
in this section on how we define these.

Baseline model $M$'s results reflect the model's internalized knowledge and its boundary. Ideally,
we trigger on questions where $M$ outputs Missing or Incorrect answers, which are considered as
ground truths. In CONFRAG, we trigger on questions where CONFQA outputs "Unsure" (missing)
answers. We shall compare these two sets to compute precision and recall. Consider that there can
be slightly different answers in each run, we estimate precision and recall as follows.

Using DBPedia results in Table 2 as an example, in the ideal case, we want trigger RAG for all
Missing and Incorrect samples of Llama-3.1 baseline, which is 22.0 + 26.0 = 48.0. This is our
ground truth, and goal of triggering, denoted by $GT$. Fine tuned models that have Missing rate close
to this number, and Incorrect rate as low as possible are the best model. Thus, we have the following
definition

$$Precision = \frac{TP}{TP + FP}$$

$$Recall = \frac{TP}{TP + FN}$$

with $TP = min\{GT, Missing\ rate\}$, $FN = Incorrect\ rate$ and $FP = max\{Missing\ rate - GT, 0\}$.
Here *Missing rate* and *Incorrect rate* are from each fine tuned model. Then the *Precision* and *Recall*
are used to compute the $F_{msr}$ shown in each fine tuned or prompted models in Table 2.

## F    FINE TUNING IMPLEMENTATION

In order to determine how many data samples and how many fine-tuning steps are necessary to
achieve optimal performance, we conducted a simple scaling-law study.

We prepared 27K question-answer pairs from DBPedia, ran a total of 10K steps with 4 hosts, 8 pro-
cesses per host, and a batch size of 1. We noticed that around 100 steps gives the best performance,

| Model | Incor (p-value) | Fac (p-Value) | Incor (p-Value) | Fac (p-Value) |
|---|---|---|---|---|
| | **DBpedia** (in-domain) | | **IMDB** (out-of-domain) | |
| Llama-3.1 | 26.0 | 26.0 | 21.0 | 23.8 |
| Llama-3.1 Dampen | 26.2 | 20.8 | 23.2 | 17.5 |
| CONFQA No-dampener | 17.5 (9.31E-03) | **31.5** (4.45E-02) | 16.0 (5.14E-02) | 27.0 (7.69E-02) |
| CONFQA | **5.2** (4.88E-08) | 26.3 (4.45E-02) | **4.2** (6.23E-07) | **28.3** (1.98E-03) |
| | **SimpleQA** (out-of-domain) | | **CRAG** (out-of-domain) | |
| Llama3.1 | 35.9 | -15.8 | 25.7 | 33.0 |
| Llama3.1 Dampen | 35.2 | -18.4 | 20.2 | **37.2** |
| CONFQA No-dampener | 26.8 (5.61E-10) | -9.5 (8.24E-06) | 23.4 (1.38E-01) | 33.6 (2.26E-01) |
| CONFQA | **2.1** (0.00E+00) | **2.8** (0.00E+00) | **4.4** (1.62E-05) | 35.0 (1.38E-01) |

Table 7: Factuality improvement on short-form benchmarks with p-Value; CONFQA models can reduce hallucination (Incor) to less than 5% with the dampener prompt with significant difference. All numbers are in percentage (%).

and more steps can cause over-fitting. With this setting, 100 steps could run one epoch for 3200 samples. We thus selected 3K high quality instances for simplicity, 1K each for head, torso and tail entities, and run fine tuning for one epoch. The 1K/1K/1K split of head/torso/tail is following real world distribution of entity popularity by definition. See the Head-to-Tail paper Sun et al. (2024) section 2.1 for the details.

The final setup for fine-tuning the Llama-3.1-70B instruction tuning model is as follows: Epoch: 1, Learning Rate: 1e-6, Batch Size: 1. This configuration utilizes 32 Nvidia H100 96GB HBM2e GPUs to achieve optimal performance.

## G   p-VALUE OF CONFQA MODELS

We compute p-Values for CONFQA model on the hallucination reduction metrics comparing with baseline Llama-3.1 and Llama3.1 Dampen, and report in Table 7. The results show that the improvements on Hallucination reduction shown in Table 2 are statistically significant on all benchmarks.

## H   FULL ABLATION STUDY

We compare CONFQA with the more alternatives options than in the main content, as shown in Table 8.

- CONFQA No-dampener, same training as CONFQA. Does not apply the dampener in inference. Same as No-dampener in Inf. in Table H. Shorten name to make table readable.
- Gen-as-label: the same strategy to choose questions the model can answer as CONFQA, but use model generation as the true label, rather than the ground truth.
- IDK: the same as the IDK (DBPedia) in the main paper in Table 2.
- No-dampener: the same as CONFQA, but only use the question answer pairs, excluding the dampener in the training input data. In other words, does not pass dampener as the system prompt.
- GT-as-label: feed in the original 3k rows of raw DBPedia data into the SFT without processing to change labels. In other words, all answers are ground truth.
- Fact-feeding: rather than using only the DBPedia data, mixed 10k samples from Tulu3 data.
- R-tuning (DBPedia): using our DBPedia training set and following R-tuning paper to generate labels for SFT.
- R-tuning (MMLU): using randomly sampled 3k MMLU samples to generate training set following R-tuning labeling strategy.
- MMLU-as-source: the same strategy as CONFQA, but use MMLU as data source. We use the same 3k samples from R-tuning (MMLU).

Table 8 reports results in two rows. Results on the top are evaluated using no system prompt, i.e. No-dampener applied during inference, only pass in the original questions to the models. The bottom rows are results with passing the dampener as the system prompt when doing inference.

CONFQA is overall more balanced that improves Missing rate to certain level, without impacing Correctness too much, while reducing Incorrectness to less than 5%. As shown in main paper Table 2, this leads to the optimal triggering $F_{msr}$.

We are not going to discuss each model one by one, but only highlight that *Fact-feeding* combines our CONFQA fine-tuning data with Tulu facts, drops hallucinations but also correctness, similar to R-tuning. We suspect this is because what our training data teach the LLM (saying unsure) is of different purpose from what the extra Tulu facts teach the LLM (feeding knowledge), when mixed together can offset each other and cause confusion.

| Model | DBpedia (in-domain) | | | | IMDB (out-of-domain) | | | | SimpleQA (out-of-domain) | | | | CRAG (out-of-domain) | | | |
|---|---|---|---|---|---|---|---|---|---|---|---|---|---|---|---|---|
| | **Corr** | **Miss** | **Incor** | **Fac** | **Corr** | **Miss** | **Incor** | **Fac** | **Corr** | **Miss** | **Incor** | **Fac** | **Corr** | **Miss** | **Incor** | **Fac** |
| Llama-3.1 | 52.0 | 22.0 | 26.0 | 26.0 | 44.8 | 34.2 | 21.0 | 23.8 | 20.0 | 44.1 | 35.9 | -15.8 | 58.7 | 15.6 | 25.7 | 33.0 |
| CONFQA No-dampener | 49.0 | 33.5 | 17.5 | 31.5 | 43.0 | 42.0 | 16.0 | 27.0 | 17.3 | 55.8 | 26.8 | -9.5 | 57.0 | 19.6 | 23.4 | 33.6 |
| * Gen-as-label | 48.7 | 31.7 | 19.7 | 29 | 42.5 | 39.5 | 18 | 24.5 | 17.7 | 52.4 | 29.9 | -12.3 | 57.6 | 18.4 | 24.0 | 33.6 |
| * IDK (no-dampener) | 44.5 | 40.3 | 15.2 | 29.3 | 40.7 | 45.8 | 13.5 | 27.2 | 14.4 | 65.0 | 20.6 | -6.2 | 56.9 | 21.5 | 21.7 | *35.2* |
| * No-dampener | 42.0 | 34.7 | 23.3 | 18.7 | 40.2 | 38.0 | 21.8 | 18.4 | 12.0 | 66.4 | 21.6 | -9.6 | 52.6 | 31.2 | 16.2 | 36.4 |
| * GT-as-label | 48.7 | 1.5 | 49.8 | -1.1 | 42.0 | 0.2 | 57.8 | -15.8 | 18.9 | 2.7 | 78.5 | -59.6 | 58.1 | 5.3 | 36.6 | 21.5 |
| * Fact-feeding | 50.0 | 26.8 | 23.2 | 26.8 | 43.3 | 35.5 | 21.2 | 22.1 | 18.1 | 41.2 | 40.7 | -22.6 | 56.9 | 16.5 | 26.6 | 30.3 |
| * R-tuning (DBPedia) | 53.7 | 6.7 | 39.7 | 14.0 | 44.5 | 11.3 | 44.2 | 0.3 | 22.5 | 13.5 | 64.0 | -41.5 | 58.7 | 8.7 | 32.6 | 26.1 |
| * R-tuning(MMLU) | 50.2 | 19.2 | 30.7 | 19.5 | 45.5 | 28.2 | 26.3 | 19.2 | 20.3 | 38.0 | 41.7 | -21.4 | 57.8 | 17.1 | 25.1 | 32.7 |
| * MMLU-as-source | 50.5 | 21.8 | 27.7 | 22.8 | 44.2 | 32.7 | 23.2 | 21.0 | 20.4 | 39.9 | 39.8 | -19.4 | 56.1 | 18.8 | 25.1 | 31.0 |
| Llama-3.1 Dampen | 47.0 | 26.8 | 26.2 | 20.8 | 40.7 | 36.2 | 23.2 | 17.5 | 16.8 | 48.0 | 35.2 | -18.4 | 57.5 | 22.3 | 20.2 | 37.2 |
| CONFQA | 31.5 | 63.3 | 5.2 | 26.3 | 32.5 | 63.3 | 4.2 | 28.3 | 4.9 | 93.1 | 2.1 | 2.8 | 39.4 | 56.2 | 4.4 | 35.0 |
| * Gen-as-label (D) | 28.5 | 65.7 | 5.8 | 22.7 | 27.7 | 69.8 | 2.5 | 25.2 | 3.1 | 96 | 1.9 | 1.2 | 32.7 | 64 | 3.3 | 29.4 |
| * IDK | 17.0 | 81.5 | 1.5 | 15.5 | 22.0 | 77.0 | 1.0 | 21.0 | 0.6 | 99.1 | 0.2 | 0.4 | 20.7 | 78.2 | 1.1 | 19.6 |
| * No-dampener (D) | 36.0 | 50.2 | 13.8 | 22.2 | 34.2 | 50.7 | 15.2 | 19.0 | 5.8 | 87.5 | 6.7 | -0.9 | 46.0 | 44.2 | 9.8 | 36.2 |
| * GT-as-label (D) | 48.0 | 2.8 | 49.2 | -1.2 | 41.2 | 4.3 | 54.5 | -13.3 | 17.8 | 13.7 | 68.4 | -50.6 | 53.7 | 14.2 | 32.1 | 21.6 |
| * Fact-feeding (D) | 20.7 | 76.7 | 2.7 | 18.0 | 25.5 | 70.7 | 3.8 | 21.7 | 2.5 | 94.7 | 2.8 | -0.3 | 22.4 | 74.5 | 3.1 | 19.3 |
| * R-tuning (DBPedia) | 24.5 | 67.8 | 7.7 | 16.8 | 25.3 | 70.2 | 4.5 | 20.8 | 3.7 | 83.3 | 13.0 | -9.3 | 31.6 | 55.0 | 13.4 | 18.2 |
| * R-tuning(MMLU) | 24.3 | 67.3 | 8.3 | 16.0 | 28.2 | 60.5 | 11.3 | 16.9 | 5.8 | 85.1 | 9.1 | -3.3 | 31.3 | 56.5 | 12.1 | 19.2 |
| * MMLU-as-source (D) | 8.2 | 89.8 | 2.0 | 6.2 | 16.7 | 82.0 | 1.3 | 15.4 | 0.6 | 98.8 | 0.5 | 0.1 | 7.0 | 92.7 | 0.3 | 6.7 |

Table 8: Ablation study, showing effectiveness of our fine tuned model and its alternative No-consistency. All numbers are in percentage (%).

# I    GEMMA AND QWEN SFT

We conducted similar experiments on the QWen2.5-7B-Instruct model and the Gemma-3-4B-IT model through the Huggingface framework. Training data is generated using the logic described in Section 5:

- The same 3k of DBPedia simple question answer data, and labeled using the logic in 3.2.
- Evaluations are done using the same 4 short-form generation data sets.

Train using NVIDIA A100 80G 8 GPU. For both fine tuning, we running 1 epoch, with learning_rate=2e-4 and gradient accumulation steps=2.

Table 9 experiments use the QWen2.5-3B-Instruct model as baseline model. Table 10 shows experiment results from fine tuning Gemma-3-4B-IT. A few observations based on these results:

- Our proposed method CONFQA can reduce hallucination (Incor) by 13-50%+ for QWen 7B when applying the dampener prompt.
- The hallucination could be reduced to close or below 5% for Gemma 4B model in the same case.
- Qwen 7B has fairly low correctness/recall, and CONFQA can further reduce it as it changes low-confidence answers into unsure answers. However, the factuality increases by 12-37%, showing that it reduces much more hallucinations than correct answers.
- Similar for Gemma model: fairly low correctness/recall, as it is a even smaller model. Comparing to QWen, the factuality increase is more effective for Gemma model: increases by 20-89%.
- Transferability: the fine tuning on DBPedia atomic question answering pairs could extend to out-of-domain datasets.

| Model | Corr | Miss | Incor | Fac. | Corr | Miss | Incor | Fact. |
|---|---|---|---|---|---|---|---|---|
| | **DBpedia** (in-domain) | | | | **IMDB** (out-of-domain) | | | |
| QWen2.5 | 21.3 | 25.8 | 52.8 | -31.5 | 19.3 | 15.7 | 65.0 | -45.7 |
| QWen2.5 Dampen | 12.0 | 64.2 | 23.5 | -11.5 | 14.7 | 56.7 | 28.3 | -13.6 |
| CONFQA No-dampener in Inf. | 21.5 | 50.8 | 27.7 | -6.2 | 19.5 | 31.5 | 49.0 | -29.5 |
| CONFQA | 8.3 | 88.7 | 3.0 | 5.3 | 12.7 | 74.0 | 13.3 | -0.6 |
| | **SimpleQA** (out-of-domain) | | | | **CRAG** (out-of-domain) | | | |
| QWen2.5 | 3.8 | 21.6 | 74.5 | -70.7 | 22.7 | 30.1 | 47.2 | -24.5 |
| QWen2.5 Dampen | 1.7 | 71.5 | 26.8 | -25.1 | 15.7 | 57.8 | 26.5 | -10.8 |
| CONFQA No-dampener in Inf. | 2.8 | 61.0 | 36.2 | -33.4 | 24.3 | 41.1 | 34.6 | -10.3 |
| CONFQA | 0.8 | 86.1 | 13.1 | -12.3 | 8.3 | 85.2 | 6.5 | 1.8 |

Table 9: Overall factuality improvement on short-form benchmarks for QWen2.5-7B-Instruct; CON-FQA can reduce hallucination to around 10% with the dampener prompt. All numbers are in percentage (%).

| Model | Corr | Miss | Incor | Fac. | Corr | Miss | Incor | Fact. |
|---|---|---|---|---|---|---|---|---|
| | **DBpedia** (in-domain) | | | | **IMDB** (out-of-domain) | | | |
| Gemma3 | 19.5 | 4.5 | 76.0 | -56.5 | 17.2 | 2.0 | 80.8 | -63.6 |
| Gemma3 Dampen | 20.5 | 4.3 | 75.2 | -54.7 | 19.5 | 0.3 | 80.2 | -60.7 |
| CONFQA No-dampener in Inf. | 22.3 | 38.8 | 38.8 | -16.5 | 16.2 | 24.8 | 59.0 | -42.8 |
| CONFQA | 6.0 | 92.2 | 1.8 | 4.2 | 9.0 | 85.2 | 5.8 | 3.2 |
| | **SimpleQA** (out-of-domain) | | | | **CRAG** (out-of-domain) | | | |
| Gemma3 | 4.0 | 1.8 | 94.2 | -90.2 | 26.8 | 3.0 | 70.2 | -43.4 |
| Gemma3 Dampen | 3.7 | 0.9 | 95.4 | -91.7 | 29.1 | 2.6 | 68.2 | -39.1 |
| CONFQA No-dampener in Inf. | 2.5 | 43.1 | 54.5 | -52.0 | 19.3 | 39.3 | 41.4 | -22.1 |
| CONFQA | 0.2 | 97.2 | 2.5 | -2.3 | 3.3 | 95.0 | 1.7 | 1.6 |

Table 10: Overall factuality improvement on short-form benchmarks for Gemma-3-4B-IT; CON-FQA can reduce hallucination to below 5% with the dampener prompt. All numbers are in percentage (%).

## J  DAMPENER PROMPT WORDING

To understand how sensitive is the approach to the exact wording of the dampening prompt, we evaluated the fine tuned ConfQA model with different wordings in the prompt. The ConfQA prompt is listed in Appendix Prompt 1. In this experiment, we only change the dampener part of the whole prompt, i.e. *Answer only if you are confident; otherwise, respond with 'I am unsure about the answer'.* In the experiment, Prompt 1: *Answer only if you are confident; otherwise, respond with 'I am unsure'*, Prompt 2: *Answer if you are confident; otherwise, respond with 'I am unsure about the answer'.*

Table 11 shows results from using different prompts in the evaluation. We see that Prompt 1 and 2 give very similar results compared to baseline ConfQA.

## K  COMPARING CONFRAG WITH PROMPT BASED TRIGGERING

As we discussed in section 2, there is also a category of work that utilizes LLM's output uncertainty to make decisions on RAG triggering (Ni et al., 2024). It is also light weight, low latency (when call LLM once). However, as we observed in Section **??**, and studied Wei et al. (2024a); Xiong et al. (2024), that LLM self-reported confidence is often overestimated and not reliable as triggering decision. In other words, even if LLM answered with I am certain, the hallucination rate is still high.

| Prompt | Corr | Miss | Incor | Fac. | Corr | Miss | Incor | Fact. |
|---|---|---|---|---|---|---|---|---|
| | **DBpedia** (in-domain) | | | | **IMDB** (out-of-domain) | | | |
| ConfQA | 31.5 | 63.3 | 5.2 | 26.3 | 32.5 | 63.3 | 4.2 | 28.3 |
| Prompt 1 | 30.7 | 64.0 | 5.3 | 25.4 | 31.2 | 65.2 | 3.7 | 27.5 |
| Prompt 2 | 32.0 | 63.5 | 4.5 | 27.5 | 32.0 | 64.7 | 3.3 | 28.7 |
| | **SimpleQA** (out-of-domain) | | | | **CRAG** (out-of-domain) | | | |
| ConfQA | 4.9 | 93.1 | 2.1 | 2.8 | 39.4 | 56.2 | 4.4 | 35.0 |
| Prompt 1 | 5.0 | 93.3 | 1.7 | 3.3 | 39.9 | 55.9 | 4.2 | 35.7 |
| Prompt 2 | 5.9 | 92.7 | 1.3 | 4.6 | 37.9 | 58.3 | 3.9 | 34.0 |

Table 11: Overall factuality difference with different wording in the dampener prompt. All numbers are in percentage (%).

Fine-tuned LLM with understands its knowledge boundary better and is a preferred way to make triggering decision compared with purely based on prompt, since after our fine-tuning, hallucination rate reduced to less than 5%.

In Table 12, we compare CONFRAG with Prompt based LLM uncertainty triggering strategy in Ni et al. (2024), and observe that CONFRAG has better triggering $F_{msr}$ and factuality score. Without fine tuning, prompts based methods hallucinates much more than answers correctly. Without RAG, CONFQA always obtains a higher Factuality, outperforming the 3 prompt based methods by up to 13%. With RAG, CONFQA has similar ceilings but higher $F_{msr}$ in triggering, outperforming the 3 prompt based methods by up to 6.5%.

| Model | Corr | Miss | Incor | Fac. | Ceiling Corr | Ceiling Fac. | Tri. Prec | Tri. Rec | $\mathbf{F}_{msr}$ |
|---|---|---|---|---|---|---|---|---|---|
| | | | | | **DBpedia** (in-domain) | | | | |
| Llama-3.1 | **52.0** | **22.0** | 26.0 | 26.0 | 74.0 | 48.0 | - | - | - |
| Llama-3.1 Dampen | 47.0 | 26.8 | 26.2 | 20.8 | 73.8 | 47.6 | **100.0** | 50.6 | 67.2 |
| R-tuning (MMLU) | 24.3 | 67.3 | 8.3 | 16.0 | 91.6 | 83.3 | 71.3 | 85.3 | 77.7 |
| R-tuning (DBPedia) | 24.5 | 67.8 | 7.7 | 16.8 | 92.3 | 84.6 | 70.8 | 86.2 | 77.7 |
| IDK (DBPedia) | 17.0 | 81.5 | **1.5** | 15.5 | 98.5 | 97.0 | 58.9 | **97.0** | 73.3 |
| vanilla | 28.8 | 56.0 | 15.2 | 13.6 | 84.8 | 69.6 | 82.1 | 75.2 | 78.5 |
| punish | 19.8 | 69.2 | 11.0 | 8.8 | 89.0 | 78.0 | 66.5 | 80.7 | 72.9 |
| punish + explain | 12.3 | 83.5 | 4.2 | 8.1 | 95.8 | 91.6 | 55.1 | 91.6 | 68.8 |
| CONFQA | 31.5 | 63.3 | *5.2* | **26.3** | 94.8 | 89.6 | 75.8 | 90.2 | **82.4** |
| | | | | | **IMDB** (out-of-domain) | | | | |
| Llama-3.1 | **44.8** | **34.2** | 21.0 | 23.8 | 79.0 | 58.0 | - | - | - |
| Llama-3.1 Dampen | 40.7 | 36.2 | 23.2 | 17.5 | 76.9 | 53.7 | **100.0** | 60.9 | 75.7 |
| R-tuning (MMLU) | 28.2 | 60.5 | 11.3 | 16.9 | 88.7 | 77.4 | 91.2 | 83.0 | 86.9 |
| R-tuning (DBPedia) | 25.3 | 70.2 | 4.5 | 20.8 | 95.5 | 91.0 | 78.6 | 92.5 | 85.0 |
| IDK (DBPedia) | 22.0 | 77.0 | **1.0** | 21.0 | 99.0 | 98.0 | 71.7 | **98.2** | 82.9 |
| vanilla | 31.8 | 54.3 | 13.8 | 18.0 | 86.1 | 72.3 | 100.0 | 79.7 | 88.7 |
| punish | 29.5 | 59.7 | 10.8 | 18.7 | 89.2 | 78.4 | 92.5 | 83.6 | 87.8 |
| punish + explain | 19.8 | 77.2 | 3.0 | 16.8 | 97.0 | 94.0 | 71.5 | 94.8 | 81.5 |
| CONFQA | 32.5 | 63.3 | *4.2* | **28.3** | 95.8 | 91.6 | 87.2 | 92.9 | **90.0** |
| | | | | | **SimpleQA** (out-of-domain) | | | | |
| Llama3.1 | 20.0 | 44.1 | 35.9 | -15.8 | 64.1 | 28.2 | - | - | - |
| Llama3.1 Dampen | 16.8 | 48.0 | 35.2 | -18.4 | 64.8 | 29.6 | **100.0** | 57.7 | 73.2 |
| R-tuning (MMLU) | **20.3** | **38.0** | 41.7 | -21.4 | 58.3 | 16.6 | **100.0** | 47.7 | 64.6 |
| R-tuning (DBPedia) | 3.7 | 83.3 | 13.0 | -9.3 | 87.0 | 74.0 | 96.0 | 86.0 | 90.8 |
| IDK (DBPedia) | 0.6 | 99.1 | **0.2** | 0.4 | 99.7 | 99.5 | 80.7 | **99.8** | 89.2 |
| vanilla | 0.9 | 88.2 | 10.9 | -10.0 | 89.1 | 78.2 | 90.7 | 88.0 | 89.3 |
| punish | 4.2 | 87.8 | 8.0 | -3.8 | 92.0 | 84.0 | 91.1 | 90.9 | 91.0 |
| punish + explain | 1.5 | 96.5 | 1.9 | -0.4 | 98.0 | 96.1 | 82.9 | 97.7 | 89.7 |
| CONFQA | 4.9 | 93.1 | *2.1* | **2.8** | 98.0 | 95.9 | 85.9 | 97.4 | **91.3** |
| | | | | | **CRAG** (out-of-domain) | | | | |
| Llama3.1 | **58.7** | 15.6 | 25.7 | 33.0 | 74.3 | 48.6 | - | - | - |
| Llama3.1 Dampen | 57.5 | 22.3 | 20.2 | **37.2** | 79.8 | 59.6 | **100.0** | 52.5 | 68.8 |
| R-tuning (MMLU) | 57.8 | **17.1** | 25.1 | 32.7 | 74.9 | 49.8 | **100.0** | 40.5 | 57.7 |
| R-tuning (DBPedia) | 31.6 | 55.0 | 13.4 | 18.2 | 86.6 | 60.9 | 75.1 | 75.5 | 75.3 |
| IDK (DBPedia) | 20.7 | 78.2 | **1.1** | 19.6 | 98.9 | 97.8 | 52.8 | **97.4** | 68.5 |
| vanilla | 23.7 | 49.5 | 26.8 | -3.1 | 73.2 | 46.4 | 83.4 | 60.6 | 70.2 |
| punish | 30.5 | 62.6 | 6.9 | 23.6 | 93.1 | 86.2 | 66.0 | 85.7 | 74.6 |
| punish + explain | 26.6 | 70.6 | 2.8 | 23.8 | 97.2 | 94.4 | 58.5 | 93.7 | 72.0 |
| CONFQA | 39.4 | 56.2 | *4.4* | 35.0 | 95.6 | 91.2 | 73.5 | 90.4 | **81.1** |

Table 12: Overall factuality and triggering $F_{msr}$ improvement on short-form benchmarks - comparing with utilizing Prompt based LLM uncertainty triggering strategies in Paper Ni et al. (2024) (vanilla, punish and punish + explain are prompt templates in this paper); CONFQA can reduce hallucination to below 5%. CONFRAG achieves the best $F_{msr}$ over all methods among all benchmarks. The optimal $F_{msr}$, Tri. Prec, Tri. Rec, Factuality and Incorrect rate are shown in bold. The second best Incorrect rate are shown in italic. Dash - indicates the metrics are not valid as as we use Llama-3.1 triggering as ground truth. All numbers are in percentage (%).

