# OpenReview forum: "ConfRAG: Confidence-Guided Retrieval-Augmenting Generation"
_ICLR.cc/2026/Conference — ICLR 2026 Conference Withdrawn Submission_

### Official Review · Reviewer_ajja · 2025-10-26

**Soundness:** 2
**Presentation:** 2
**Contribution:** 2
**Rating:** 2
**Confidence:** 3

**Summary:**

This paper introduces a ConfRAG, a RAG strategy that only triggers retrieval when the answer has low confidence. To implement this strategy, the authors introduce a fine-tuning method, ConfQA, which teaches the LLM to state “I am unsure” when the answer is incorrect, via a dampening prompt “Answer only if you are confident” and a training dataset composed of atomic factual statements.

**Strengths:**

The proposed finetuning method can teach LLM to refrain from generating inconfident outputs while simultaneously improving retrieval efficiency.

**Weaknesses:**

Clarity
- The paper's central premise is to use model uncertainty as a trigger for retrieval. However, there appears to be a fundamental mismatch between this trigger and the fine-tuning objective, which is based on correctness. The paper does not sufficiently address the gap between model confidence and answer correctness. For instance, a model can be uncertain about a correct answer or highly confident in an incorrect one. This discrepancy seems to undermine the core mechanism, and it is unclear how the proposed method accounts for it.

Novelty
- The paper's claimed contributions for RQ1 (confidence calibration) and RQ2 (encouraging abstention) appear to substantially overlap with existing literature. The overconfidence of LLMs [1-2] and the use of "I don't know" for selective abstention [3-5] are both well-studied topics. The authors should more clearly articulate the specific novelty of their approach in light of this extensive prior work. As written, the incremental contribution is not clear.

Experiments
- The empirical evidence is not compelling. ConQA seems to significantly harm correctness on short-form benchmarks, while the performance improvements on long-form generation tasks are marginal. For instance, in Table 5, the proposed method is consistently outperformed by a standard RAG baseline with contriever.

- The paper fails to compare against standard methods for hallucination detection (e.g., P(True) [6], semantic uncertainty [7]) or abstention (e.g., [8,9]). Adaptive RAG baselines, such as self-RAG and DRAGIN, were mentioned in the related work section but not compared experimentally.

[1] Xiong, Miao, et al. "Can LLMs Express Their Uncertainty? An Empirical Evaluation of Confidence Elicitation in LLMs." The Twelfth International Conference on Learning Representations.
[2] Tian, Katherine, et al. "Just ask for calibration: Strategies for eliciting calibrated confidence scores from language models fine-tuned with human feedback." arXiv preprint arXiv:2305.14975 (2023).
[3] Cheng, Qinyuan, et al. "Can AI assistants know what they don't know?." arXiv preprint arXiv:2401.13275 (2024).
[4] Chen, Lida, et al. "Teaching large language models to express knowledge boundary from their own signals." arXiv preprint arXiv:2406.10881 (2024).
[5] Li, Jiaqi, Yixuan Tang, and Yi Yang. "Know the unknown: An uncertainty-sensitive method for llm instruction tuning." arXiv preprint arXiv:2406.10099 (2024).
[6] Kadavath, Saurav, et al. "Language models (mostly) know what they know." arXiv preprint arXiv:2207.05221 (2022).
[7] Kuhn, Lorenz, et al. "Semantic Uncertainty: Linguistic Invariances for Uncertainty Estimation in Natural Language Generation." The Eleventh International Conference on Learning Representations.
[8] Feng, Shangbin, et al. "Don't hallucinate, abstain: Identifying LLM knowledge gaps via multi-LLM collaboration." arXiv preprint arXiv:2402.00367 (2024).
[9] Yadkori, Yasin Abbasi, et al. "Mitigating llm hallucinations via conformal abstention." arXiv preprint arXiv:2405.01563 (2024).

**Questions:**

Please refer to the weaknesses above.

---

> ### Author Response · Authors · 2025-11-26
>
> We would like to express our sincere gratitude for your time, effort, and invaluable feedback during the review process of our paper. Thank you for your careful reading and for sharing your expertise. To address your thoughtful comments and questions, we present responses to each of your questions/concerns in the following section and hope it can clarify your concerns.
>
> 1. Clarity
>
> Answer: We fine tuned the model to only answer a question if it can answer the question correctly. Table 2 shows that when the fine-tuned model has a high confidence and gives an answer, the correctness is high and the hallucination rate is very low.
>
> 2. Novelty:
>
> Answer: The paper makes two major contributions:
>
> a. ConfRAG as a RAG triggering strategy that utilize ConfQA as a triggering decision maker.
>
> b. ConfQA as a fine-tuned model to decide when to answer user questions directly, or should consult external knowledge, aka RAG.
>
> We thank the reviewer for the references. We have revised the Related Work session and added the references. We next briefly discuss them as follows.
> - [1] proposed an evaluation of methods for eliciting and measuring uncertainty in LLMs. Very similar to the SimpleQA paper [a] we cited.
> - [2] proposed to ask the model to generate alternative answers before stating a confidence score  which then significantly improves the model self reported calibration. Although the method is effective, it is not applicable in real applications due to increased latency.
> - [3] is cited in our paper in line 79 & 148, which requires consistency, thus is similar to IDK. We compared and reported in Table 2, showing that requiring consistency can unnecessarily drop accuracy.
> - [4] proposed a two stage unsupervised training process,
> - [5] focuses on constructing contextual QA training data,  including a scenario where the provided context is intentionally insufficient. This work is orthogonal to our work focusing on RAG triggering.
>
> 3. Experiments:
>
> Answer to first one:
>
> By teaching an LLM model not to give a low-confidence answer, we inevitably tradeoff hallucinations w. missing answers. We think our solution is superior for three reasons.
>
> a. increased factuality (corr%-incorr%) by up to 20%: turned much more hallucinations than correct answers to “unsure”.
>
> b. recommended that if a high missing rate is undesired, one can apply ConfQA w/o dampener in inference, maintaining similar accuracy while dropping hallucination by up to 10%.
>
> c. showed that when using ConfQA for RAG triggering, QA accuracy could increase to >95%, achieving high accuracy and reducing retrievals.
>
>
> Answer to second one:
>
> - Paper [6] was cited in our paper (Line 118), the proposed method is in a pre-training setup, opposed to our post training SFT. Thus we didn’t do any comparison as they are not comparable.
> - Paper [7] introduces "semantic entropy," which measures uncertainty at the level of semantic meaning rather than individual word sequences. This is different from our goal, which measures confidence in the correctness of a value.
> - [8,9] propose multi-LLM collaboration or ask LLM to generate responses multiple times,  and then to use compete/consensus to identify LLM knowledge gaps and calibration with confidence.
> - Self-RAG (cited in Line 105 already) proposes to trigger RAG multiple times. Triggering RAG multiple times increases latency and is expensive, not suitable for RAG triggering. We compared IDK, which leverages consistency in fine-tuning (Table 1).

---

### Official Review · Reviewer_TJwu · 2025-10-27

**Soundness:** 2
**Presentation:** 3
**Contribution:** 2
**Rating:** 2
**Confidence:** 4

**Summary:**

This paper focuses on an important question: when should an LLM trigger retrieval. It first measures the knowledge boundaries of the LLM, then uses correctness annotations as supervision to train the model to express uncertainty when it does not know the answer, and performs retrieval only when the model expresses uncertainty.

**Strengths:**

1.  The paper focuses on an important question: teaching the model to recognize its own knowledge boundaries and to trigger retrieval only when it does not know the answer.
2. The paper is well written and logically coherent.
3. The paper uses the principle of “answer only if you are confident” to suppress overconfidence, and it trains on atomic facts, which leads to relatively high accuracy.

**Weaknesses:**

1. The paper lacks novelty — the idea of triggering retrieval only when the model does not know the answer is not new.
2. There have been many works between 2023 and 2024 that use SFT (Supervised Fine-Tuning) to enable models to express uncertainty, and this paper is not fundamentally different from those approaches.
3. The paper includes too few baselines and lacks citations to several foundational works in the areas of adaptive RAG and LLM knowledge boundary perception.


[1] SAC3: Reliable Hallucination Detection in Black-Box Language Models via Semantic-aware Cross-check Consistency. EMNLP 2023

[2] When Do LLMs Need Retrieval Augmentation? Mitigating LLMs' Overconfidence Helps Retrieval Augmentation. ACL 2024

[3] Self-RAG: Learning to Retrieve, Generate, and Critique through Self-Reflection. ICLR 2024

[4] Alignment for Honesty. NeurIPS 2024

[5] Teaching Models to Express Their Uncertainty in Words. TMLR 2022

**Questions:**

See Weaknesses.

---

> ### Author Response · Authors · 2025-11-26
>
> We would like to express our sincere gratitude for your time, effort, and invaluable feedback during the review process of our paper. Thank you for your careful reading and for sharing your expertise. To address your thoughtful comments and questions, we present responses to each of your questions/concerns in the following section and hope it can clarify your concerns.
>
> 1. The paper lacks novelty — the idea of triggering retrieval only when the model does not know the answer is not new.
>
> Answer: The given references [2][3] are focusing on how to utilize retrieved doc to generate truthful responses, i.e. retrieval has already happen, then model learns to say I don’t know if retrieved doc is insufficient, or lies in the category of triggering RAG multiple times in which cases increases latency. Instead, we focus on deciding whether to trigger RAG.
> We are the first one to propose a very simple and reliable way to train LLMs to reduce hallucination to < 5%, and the first one to propose combining LLMs hallucination reduction and RAG triggering. The final proposed application architecture achieves lower latency while maintaining accuracy.
>
> 2. There have been many works between 2023 and 2024 that use SFT (Supervised Fine-Tuning) to enable models to express uncertainty, and this paper is not fundamentally different from those approaches.
>
> Answer: Although the idea of using SFT to reduce hallucination or enable models to respond with confidence is not new, we are the first to show that the hallucination rate can be diminished to <5%, and thus reducing RAG triggering but still achieving high QA accuracy. The two critical ingredients for hallucination reduction–dampener prompt in training and using atomic factual questions–have not been proposed elsewhere.
>
> 3. The paper includes too few baselines and lacks citations to several foundational works in the areas of adaptive RAG and LLM knowledge boundary perception.
>
> Answer: We thank the reviewer for the references and have revised the Related Work section with new citations. We have conducted extensive experiments and compared with work closest to ours, such as R-tuning and IDK. Among these citations
>
> - [1] and [4] requires consistency, thus is similar to IDK. We compared and reported in Table 2, showing that requiring consistency can unnecessarily drop accuracy.
> - [2][3] requires always-RAG, thus causing higher latency, as discussed in the response to W1.
> - [5] similar to [1][4] and IDK, and it is the  first time a model has been shown to express calibrated uncertainty about its own answers in natural language.

---

> > ### Comment · Reviewer_TJwu · 2025-11-28
> > **Reviewer Response**
> >
> > I think your response contains factual errors. For example, [2] determines whether to trigger retrieval before the retrieval actually occurs.
> >
> > I feel that training model confidence in an SFT manner and then deciding whether to trigger retrieval overlaps heavily with existing work, which aligns with what Reviewer ajja mentioned. I recommend conducting a thorough literature review before writing the paper and avoid claiming your work as “the first” too easily.
> >
> > I decide to keep my score.

---

> > > ### Author Response · Authors · 2025-12-03
> > > **New benchmark compared with paper [2]**
> > >
> > > Thank you so much for pointing out this. Sorry we miss-understood paper [2]. Paper [2] prompts the LLM to output uncertainty, and triggers RAG only if the LLM outputs “uncertain” together with the response. We updated our paper accordingly in the latest revision.
> > >
> > > We also did an experiment to compare its triggering strategy precision / recall / F_msr with our proposed method. Our method is superior compared to the prompt based methods proposed in paper [2]. Please see the results in Table 12, and the discussion in Appendix K. To summarize:
> > >
> > > - Without RAG, ConfQA always obtains a higher Factuality, outperforming [2] by up to 13%.
> > > - With RAG, ConfQA has similar ceilings but higher F-msr in triggering, outperforming [2] by up to 6.5%.

---

### Official Review · Reviewer_2uqx · 2025-11-01

**Soundness:** 2
**Presentation:** 2
**Contribution:** 2
**Rating:** 2
**Confidence:** 3

**Summary:**

This work focuses two key challenges in factual question answering with Large Language Models (LLMs): reducing hallucinated answers and minimizing unnecessary retrieval operations in Retrieval-Augmented Generation (RAG) systems.
The authors introduce ConfQA, a fine-tuning method where the model is trained to answer only when confident and otherwise respond with “I am unsure.”
This is achieved via a dampening prompt and training on atomic factual statements.
Building upon this, they propose ConfRAG, a system that triggers RAG only if the base model expresses uncertainty, resulting in reduced hallucination rates and fewer external retrievals, while maintaining high accuracy.

**Strengths:**

1. The motivation is clear. This work focuses important issues of LLM hallucinations and computational efficiency in RAG.
2. The use of confidence signaling (“I am unsure”) and SFT objectives makes the framework conceptually straightforward.
3. Experimental results demonstrate improvements over baselines across multiple benchmarks, notably lowering hallucination rates.

**Weaknesses:**

1. The primary limitation of this work is its lack of novelty.
Training with the 'unknown' token is a commonly employed technique in many existing RAG systems (e.g., [1]).
This study does not offer substantial new insights beyond some empirical observations.

2. Several design choices are not clearly explained. For example, the rationale behind the design of the dampener prompt and the "unsure" answer remains unclear. Is model performance highly sensitive to the choice prompt and answer? Additionally, how to adjust the proportion of "unknown" answers in the training data, and is this ratio critical to overall model performance? Regarding training with negative signals, what is the justification for using SFT instead of DPO or PPO? Providing further details and analysis on these points would strengthen the contribution of this work.

[1] Enhancing Retrieval-Augmented Generation with Dehallucinating Parallel Context Extension.

**Questions:**

See weakness.

---

> ### Author Response · Authors · 2025-11-26
>
> We would like to express our sincere gratitude for your time, effort, and invaluable feedback during the review process of our paper. Thank you for your careful reading and for sharing your expertise. To address your thoughtful comments and questions, we present responses to each of your questions/concerns in the following section and hope it can clarify your concerns.
>
> 1. The primary limitation of this work is its lack of novelty. Training with the 'unknown' token is a commonly employed technique in many existing RAG systems (e.g., [1]). This study does not offer substantial new insights beyond some empirical observations.
>
> Answer:  We note that our work is fundamentally different from training RAG systems with ‘unknown” tokens to avoid hallucinations from retrieval noises, as [1] does. Instead, our goal is to decide if the LLM model has internalized knowledge to answer a given question reliably, or need to conduct web search.
>
> The key contribution of our approach is an effective RAG triggering strategy. In particular, we are the first to show that a fine-tuned model can reduce hallucination rate in QA <5%, with two critical ingredients for hallucination reduction—dampener prompt in training and use of atomic factual questions. This model allows us to trigger RAG only when necessary but still achieves high QA accuracy.
>
> 2. Several design choices are not clearly explained. For example, the rationale behind the design of the dampener prompt and the "unsure" answer remains unclear. Is model performance highly sensitive to the choice prompt and answer? Additionally, how to adjust the proportion of "unknown" answers in the training data, and is this ratio critical to overall model performance? Regarding training with negative signals, what is the justification for using SFT instead of DPO or PPO? Providing further details and analysis on these points would strengthen the contribution of this work.
>
> Answer: The dampener prompt aims to instruct the model to refrain from answering a question based on its confidence. Our experiments (Table 2) verified that this instruction indeed plays the role.
>
> Experiments with different prompt wording shows that model performance remains similar within the confidence interval. We will add these results in the next revision.
> For different answers to the same question, we have tried using both ground truth (GT-as-label), or model generated correct response (Gen-as-label) as true label , and did ablation study (results shown in Table 8 in appendix), which shows GT-as-label performs slightly better than using model generated correct response.
>
> Adjust the proportion of “unknown” answers: we have studied different ways of selecting “unknown” answers including different portions based on how many times the model can answer a question correctly. Within a range, SFT is stable (our paper proposed to ask the model to answer questions once, and identify questions model answered correctly as positive sample, i.e. ground truth as label, and questions model answered incorrecting or not answered as negative sample, i.e. I am unsure as the label). This is stable (reproducible). In extreme cases, for example, we require a question to be answered correctly 10 out of 10 times (IDK paper) to select it as positive sample, the reduction of hallucination as well as correctness could be significant.
>
> For DPO or PPO, it has been studied in the IDK paper. We mentioned this as our future work in the Limitation section.

---

### Official Review · Reviewer_7owA · 2025-11-01

**Soundness:** 3
**Presentation:** 3
**Contribution:** 3
**Rating:** 6
**Confidence:** 2

**Summary:**

This paper introduces ConfRAG, a framework for triggering Retrieval-Augmented Generation (RAG) based on model confidence. The authors propose ConfQA, a fine-tuning strategy that teaches LLMs to respond with "I am unsure" when they lack confidence in factual answers. The key contributions include: (1) a RAG triggering mechanism based on explicit confidence assessment, (2) a fine-tuning method using atomic factual statements from DBPedia with a "dampening prompt," and (3) comprehensive evaluation across 7 benchmarks showing hallucination reduction to <5% while maintaining accuracy comparable to always-on RAG with reduced latency.

**Strengths:**

- The paper methodically addresses the RAG triggering problem with clear motivation (Figure 3 shows LLMs are overconfident)
- Strong empirical results:

1. Consistent hallucination reduction across diverse benchmarks
2. Maintains or improves factuality scores
3. Practical latency improvements demonstrated


- The dampening prompt and DBPedia focus are well-motivated through ablations
- 7 benchmarks covering different question types and domains show generalization
- Detailed prompts, implementation details, and clear methodology facilitate reproduction
- The framework is lightweight (no hidden-state inspection) and compatible with existing systems

**Weaknesses:**

- The core contribution beyond R-Tuning and IDK appears to be (1) the dampening prompt and (2) using DBPedia instead of MMLU. While effective, this feels incremental.
- Training data limitations:

1. Only 3K samples seems small for teaching general confidence behavior
2. DBPedia focus on entity attributes may not transfer well to other factual question types
3. No systematic study of data diversity vs. quality trade-offs


- Fine-tuning results are primarily on Llama-3.1-70B. Claims about generalization would be stronger with results across model families and sizes.
- RAG baseline concerns:

1. The "ideal RAG" assumption (Section 3.1) is unrealistic
2. Real RAG results (Table 3) show modest improvements, and on CRAG, RAG-everywhere still underperforms LLM-only
3. More sophisticated RAG methods should be compared


- Evaluation gaps:

1. No human evaluation to validate LLM-as-a-judge decisions
2. The "ceiling" metrics assume perfect RAG, which may be misleading
3. Missing analysis of failure modes (when does ConfQA incorrectly say "unsure"?)



- Limited theoretical insight: Why does the dampening prompt work so well? Why does DBPedia generalize better than MMLU? The paper is primarily empirical without deeper understanding.

**Questions:**

- Can ConfQA fine-tuned on Llama-3.1-70B be used to trigger RAG for other models (e.g., GPT-4o)? Or does each model need separate fine-tuning?
- How sensitive is the approach to the exact wording of the dampening prompt? Have you experimented with variations?
- What is the optimal mix of entity popularities (head/torso/tail) in training data? Current 1K/1K/1K split seems arbitrary.
- In Table 2, ConfQA still has 5.2% incorrect on DBPedia (in-domain). What characterizes these failures? Are they specific entity types or question patterns?
- How do you determine if a question requires dynamic information? This seems crucial for the architecture in Figure 2.
- How does this compare to calibration techniques or uncertainty estimation methods that don't require fine-tuning?
- What is the cost of fine-tuning vs. the savings from reduced RAG calls? Is this approach cost-effective at scale?
- he long-form results (Table 5) show modest improvements. Why doesn't the approach transfer as well to multi-claim scenarios?
- Is binary "unsure" vs. answer optimal? Have you explored soft confidence scores that could enable more nuanced triggering?
Real-world deployment: Have you deployed this in production? What practical challenges emerged beyond the experimental setting?

---

> ### Author Response · Authors · 2025-11-26
>
> We would like to express our sincere gratitude for your time, effort, and invaluable feedback on our paper. To address your thoughtful comments and questions, we present responses to each of your questions/concerns in the following section and hope it can clarify your concerns.
>
> Q1: Each LLM model needs to be fine-tuned, as their knowledge boundary might be different. Thus this approach is not applicable for black-box models (discussed in section 7)
>
> Q2: Added experiment results in the appendix J
>
> Q3: The split is following real world distribution of entity popularities by definition. See the Head-to-Tail paper section 2.1
>
> Q4: The 5.2% incorrect responses contain 18 head, 6 torso, 7 tail entities. The failures regarding head questions are mostly due to the answers to the questions being not unique or ambiguous.
>
> Q5: We fine-tune a Llama-3.2-8B with precision 93, recall 85 to determine if a question is a dynamic question or static. How to best train this model is out of the scope of this paper
>
> Q6: There are two major methods for uncertainty estimation without fine-tuning: 1). applies prompt engineering to ask LLMs to self-report its confidence. As Section 4.2 shows, these methods are often over-confident. 2). requires calling LLMs multiple times to decide confidence. Although the confidence is better calibrated (Appendix D), the method is not practical due to high latency/cost
>
> Q7: We would like to refrain from always-RAG because of the latency increase in generating responses (>1s). Fine-tuning is offline so does not incur latency. Additionally, we use 3k samples and train 1k steps to get decent performances
>
> Q8: For long-form questions, we did not apply the dampener but asked the model to provide specific details (Ln 468-470). Similar to observation for short-form answers, no dampening in inference will improve precision only moderately (Table 4)
>
> Q9: We did a quick experiment on CRAG, with 5 confidence levels. We apply threshold to trigger RAG for low /mid (level 1-4) confidences. With this, we trigger RAG for 47% of CRAG questions. For the 53% of questions that LLM has high confidence, model only has 67% accuracy. Wtih perfect RAG, the ceiling correctness is 82.5%. Much lower than 95.6%(using ConfQA in Table 2 in our paper).
> We are sorry, the deployment status is confidential. However, the architecture design is straightforward and we don’t see major practical challenges.
>
> Weaknesses:
>
> W1: We are the first one to propose a simple and reliable way to train LLMs to reduce hallucination to < 5%, and the first one to propose combining LLMs hallucination reduction and RAG triggering. The final proposed application architecture achieves lower latency while maintaining accuracy
>
> W2: 1). In Appendix F, we briefly described the reason we chose 3k samples. 2). We evaluated the ConfQA model also on IMDB, CRAG, SimpleQA (Table 2) and long form generation (Table 5), with CRAG covering complex factual questions, and long form gen covering multi-claims longer generation, and saw the transferability of our proposed method. 3).  Ablation on data sources are in Table 4, and full ablation study is in Table 8. Other data sources we tried include MMLU and Tulu3
>
> W3: we added results from QWen and Gemma to Appendix I
>
> W4: 1). We use the “Ideal RAG” assumption as a simplicity for easy discussion. In Section 5.3, we show in Table 2 the real implementation of RAG triggering that gives the best Fmsr. Table 3 shows the real world RAG implementation accuracy and latency.
> 2). Table 3 shows improvement of ConfRAG on both correctness and factuality, although not substantial, the latency has dropped a lot compared with RAG-everywhere.
> 3). ConfRAG approach advantage: simple to implement (compared to token level confidence based triggering) and does not add to latency (compared to adaptive RAG or always trigger RAG).
>
> W5: 1). LLM-as-a-judge is widely used and we cited this paper https://arxiv.org/abs/2406.04744 in Ln 246 that contains information vetted by humans. We didn’t include the details in the paper as the method we use has been proved in several papers, and also it’s not our focus.
> 2).  a). Improving RAG is outside of the scope of this paper. This study is independent of RAG implementation. b).For triggering decisions, we assume perfect RAG. Reasoning is when a model cannot answer a question successfully, we rely on RAG, and leave RAG accuracy as a separate topic. Table 3 shows the real world RAG implementation accuracy and latency
> 3). See above answer.
>
> W6: Dampener prompt works well as it uses the model’s instruction following capability.
> DBpedia generalizes better: 1).contains atomic facts, building blocks for complex factual information, so easier to help to align on a model's confidence. 2). The head/torso/tail even split resembles the real world distribution of entity types as defined in the Head-to-Tail paper section 2.1.
> MMLU mixes facts with reasoning data and can introduce ambiguity, discussed in paper (Ln 373-399).

---

### Note · Authors · 2026-01-06

I have read and agree with the venue's withdrawal policy on behalf of myself and my co-authors.